# The Effects of a Weight-Loss Herbal Formula RCM-107 and Its Eight Individual Ingredients on Glucagon-Like Peptide-1 Secretion—An In Vitro and In Silico Study

**DOI:** 10.3390/ijms21082854

**Published:** 2020-04-19

**Authors:** Shiqi Luo, Harsharn Gill, Bryce Feltis, Andrew Hung, Linh Toan Nguyen, George Binh Lenon

**Affiliations:** 1School of Health and Biomedical Sciences, RMIT University, PO Box 71, Bundoora West Campus, Victoria 3083, Australia; s3369593@student.rmit.edu.au (S.L.); bryce.feltis@rmit.edu.au (B.F.); 2School of Science, RMIT University, GPO Box 2476, Melbourne, Victoria 3001, Australia; harsharn.gill@rmit.edu.au (H.G.); andrew.hung@rmit.edu.au (A.H.); 3Department of Endocrine, 103 Military Hospital, Vietnam Military Medical University, Hanoi 100000, Vietnam; toannl@vmmu.edu.vn

**Keywords:** GLP-1 secretion, appetite suppression, obesity, overweight, cell culture, in silico analysis, herbal medicine

## Abstract

Obesity is a multifactorial disease that can lead to other health issues. Glucagon-like peptide-1(GLP-1), as one of the satiety signal, has been linked with appetite suppression and weight loss. Due to the limitations of GLP-1 and its analogues, alternative treatments such as herbal therapies have become popular. The herbal formula RCM-107 has demonstrated its inhibitory effects on lipid and carbohydrate absorption in our previous work. However, no published data described its effects on GLP-1 secretion. Therefore, this study aimed to determine the effects of RCM-107 and its individual ingredients on GLP-1 secretion via enzyme-linked immunosorbent assay (ELISA). Furthermore, molecular docking was performed to predict the key chemical compounds that are likely to be GLP-1 receptor agonists. *Gardeniae fructus*, one of the ingredients in RCM-107, demonstrated significantly greater effects on inducing GLP-1 secretion than the positive control epigallocatechin gallate (EGCG). Two *Gardeniae fructus* ligands, 3-epioleanolic acid and crocin were predicted to bind to the active form of GLP-1 receptor at the binding pocket with residues known for the receptor activation, suggesting that they could potentially serve as receptor agonists. Overall, this study reported the effects of researched herbs on GLP-1 secretion and proposed two compounds that may be responsible for antiobesity via GLP-1 receptor activation.

## 1. Introduction

Obesity is considered a complicated multifactorial disease. Globally, almost one third of the world population is classified as overweight or obese [1]. Obesity has negative impacts on physiological functions, causing a threat to public health. It increases the risk of developing metabolic syndromes such as diabetes mellitus, cardiovascular disease and several types of cancers [1,2,3]. Even though the prevalence of obesity has increased in both sexes at all ages with different geographical locality, ethnicity or socioeconomic background, older people and women seem to have greater obesity rates [1].

There are various approaches to reduce weight, including decreasing nutrition absorption or suppressing appetite [4]. Glucagon-like peptide-1(GLP-1) is one of the “satiety signals’’ that reduces hunger. It is known as an incretin hormone, secreted by enteroendocrine L cells in the gastrointestinal tract (GI) [5]. The amount of GLP-1 secretion is in response to calory intake. It can be stimulated by both fats and carbohydrates, while protein has only a minor impact [6]. The existing two major molecular forms of GLP-1 include GLP-1 (7-36) and GLP-1 (7-37), while the circulating active GLP-1 is mainly found in the form of GLP-1(7-36) [7].

Reports showed that native GLP-1 has a very short half-life (less than 2 min) in vivo as a result of degradation by dipeptidyl peptidase-4 (DPP-4) in plasma [7,8]. Liraglutide, a stable GLP-1 analogue, was approved by FDA for treating obesity in 2014. It can suppress appetite and increase satiety by activating the GLP-1 receptor, which is located in the hypothalamus, gastrointestinal tract and pancreas. However, it is a daily injectable treatment that is also relatively high cost [6,9]. 

Due to these limitations on GLP-1 and its analogues, cost-effective herbal supplements, including natural products, medicinal plant extracts and Chinese herbal medicine, are popular alternatives for weight reduction. Natural products such as epigallocatechin gallate (EGCG) [5], geniposide [10], berberine [11] and ginsenoside metabolite compound K [12] have been shown to stimulate GLP-1 secretion in in vitro experiments. 

In this paper, we have investigated the commercially available RCM-107 formula (Slimming Plus), a modification of our previously studied RCM-104 formula, which demonstrated significant effects on weight loss in clinical trials [13]. RCM-107 contains eight Chinese herbs, including *Camellia sinensis*/Lu cha ye; *Poria*/*Poria cocos (schw.) Wolf.*/Fu ling; *Nelumbinis folium*/*Nelumbo nucifera Gaertn.*/He ye; *Alismatis rhizome*/*Alisma orientalis (sam.) Juzep.*/Ze xie; *Plantaginis semen*/*Plantago asiatica L.*/Che qian zi; *Cassiae semen*/*Cassia obtusifolia L.*/Jue ming zi; *Sophorae flos*/*Sophora japonica L./*Huai hua; and *Gardeniae fructus*/*Gardenia jasminoides Ellis.*/Zhi zi [14]. Our recent work has demonstrated the potent pancreatic lipase inhibitory effect of this formula, which can lead to decreased lipid absorption and, thus, potentially weight loss [4]. In addition, we have preliminary data suggesting that there are amylase inhibition effects of this formula that reduce carbohydrate digestion. We have also identified the presence of six active chemical compounds in RCM-107 and related single herbs, such as EGCG, epicatechin-3-gallate (ECG), (−)-epicatechin (EC), caffiene, rutin and crocin [4]. However, to our knowledge, no published scientific data describes the effects of RCM-107 and its eight ingredients on GLP-1 secretion.

Given this lack of information, this present study aims to firstly determine the effects of RCM-107 and its individual ingredients on GLP-1 secretion in NCI-H716 cells using the enzyme-linked immunosorbent assay (ELISA). A variety of animal or human cellular models, such as mouse intestine cell lines (STC-1), human colorectal cells (NCI-H716) and murine endocrine cells (GLUTag), have been used to investigate satiety hormone secretion [5]. The NCI-H716 cell line (intestinal enteroendocrine L-cells) is the only human model used for in vitro experiments to study GLP-1 regulation. It is derived from the ascites fluid of a 33-year old Caucasian male with colorectal adenocarcinoma. Furthermore, the key chemical compound(s) that may potentially act as a GLP-1 receptor agonist will be proposed. 

## 2. Results

### 2.1. Cell Viability

The results (Figure 1a,b) showed cell viability (%) of NCI-H716 cells in the presence of EGCG, the RCM-107 formula and its eight individual herbal components at various concentrations over 24 h. A progressive decline in cell viability was observed, which indicated that the majority of the test samples became toxic to NCI-H716 cells in a dose-dependent manner. Among all samples, EGCG and Camellia sinensis presented the most significant decrease in cell viability with less than 10% viable cells at 1 mg/mL compared to untreated cells (*p* < 0.0001). In addition, RCM-107 became cytotoxic at concentrations of 0.05 mg/mL and greater (*p* < 0.01), leading to a cell viability less than 15% at the highest concentration (*p* < 0.0001). 

Nelumbinis folium demonstrated statistically significant toxicity at a concentration of 0.1 mg/mL with 51% viability at the highest concentration (*p* < 0.0001), while Gardeniae fructus showed statistically significant toxicity from the concentration 0.5 mg/mL with 63% viability at the highest concentration (*p* < 0.0001). Similarly, Sophorae flos showed significant toxicity only at or above the concentration of 0.5 mg/mL, with 62% viability at the highest concentration (*p* < 0.0001). 

On the other hand, Alismatis rhizome showed the least toxicity with 99% viability even at the highest concentration (*p* > 1). Interestingly, this herb significantly increased the cell viability at a concentration of 0.05 and 0.1 mg/mL (*p* < 0.0001; *p* < 0.005). Cassiae semen was less toxic to cells until the concentration reached 1 mg/mL, which led to 88% viability (*p* < 0.0001). Both Poria and Plantaginis semen showed significant cytotoxicity when the concentration was higher than 0.8 mg/mL with 84% viability at the highest concentration (*p* < 0.0001; *p* < 0.0001). 

This toxicity data indicated that EGCG, Camellia sinensis and RCM-107 were more cytotoxic to NCI-H716 cells than the other tested herbal samples.

### 2.2. GLP-1 Secretion

The effect of RCM-107 and its eight ingredients on the protein expression of GLP-1 was quantified using ELISA. Cells were treated with herbal samples and the positive control EGCG for 2 h. The maximal concentrations used for ELISA were selected from the highest concentration that maintained cell viability above 96% (0.01 mg/mL RCM-107, Camellia sinensis, Gardeniae fructus and EGCG, 0.05 mg/mL Nelumbinis folium; 0.1 mg/mL Sophorae flos; 0.5 mg/mL Plantaginis semen; 0.8 mg/mL Poria, Cassiae semen; and 1 mg/mL Alismatis rhizome). Sophorae flos demonstrated the most significant increase in releasing GLP-1 (1.5328 ng/mL) compared to untreated cells (*p* < 0.0001), followed closely by Gardeniae fructus (1.4688 ng/mL) and Alismatis rhizome (1.4424 ng/mL) (*p* < 0.0001). In addition, Nelumbinis folium (1.3516 ng/mL), Cassiae semen (1.2662 ng/mL) and Poria (1.2365 ng/mL) also led to a statistically significant increase of GLP-1 secretion (*p* < 0.0001). The RCM-107 formula (1.1970 ng/mL) and Camellia sinensis (1.1954 ng/mL) showed increased GLP-1 release; however, they were not statistically significant (*p* > 0.5). On the other hand, Plantaginis semen significantly reduced the amount of GLP-1 (*p* < 0.0001) (Figure 2). 

The positive control EGCG (1.2906 ng/mL) also significantly increased GLP-1 (*p* < 0.0001); however, Gardeniae fructus had greater effects on GLP-1 secretion than this positive control when they were assayed at the same concentration (*p* < 0.0001) (Figure 2). 

Even though Sophorae flos exhibited the highest protein secretion level, the concentration used in the experiment was 10 times higher than Gardeniae fructus due to its lower cytotoxicity to NCI-H716 cells. Therefore, Gardeniae fructus is considered to be more vital than other single herbs in RCM-107 for GLP-1 stimulation. 

### 2.3. Molecular Docking

In order to understand the possible GLP-1 agonist-mediated receptor activation mechanisms, 22 compounds from Gardeniae fructus were docked with the target protein described by Protein Data Bank identification code (PDBID) 5NX2 (agonist-bound active conformation of GLP-1) (Table 1). According to our results, the top three compounds with docking energies superior (more negative) to the positive control EGCG (*−*8.4 kcal/mol) include GF1 stigmasterol (*−*10.9 kcal/mol) followed by GF2 3-epioleanolic acid (*−*10.8 kcal/mol) and GF3 crocin (*−*10.7 kcal/mol). To have a deeper knowledge of the potential role of the above three compounds, the three-dimensional (3D) ribbon diagrams and two-dimensional (2D) ligand-target interactions were inspected (Figure 3 and Figure 4). GF2 and GF3 formed hydrogen bonding interactions with amino acid residues such as Lys197, Lys202, Glu69, Arg121, Arg215, Asp198 and Tyr148, on 5NX2, while EGCG also formed the same type of interaction with Asp198 and Tyr148 (Table 2). GF1 however did not form any hydrogen bonds with amino acid residues on the GLP-1 receptor. 

Other types of interactions such as such as π-Alkyl and π-σcontacts have been observed. GF1 formed multiple hydrophobic interactions with residues like Trp39, Leu32, Val36 and Met204, while GF2 had multiple hydrophobic contacts with Leu142, Lys202, Tyr205, Pro137 and Leu201, out of which EGCG also formed this type of interaction with Leu201. GF3 only formed single hydrophobic interactions with residues Tyr205, Leu32 and Trp33 (Figure 4). 

Thus, both GF2 and GF3 form hydrogen bonds with a number of residues at the predicted binding site of GLP-1, some of which are also bound by the positive control EGCG. Amongst the ligands studied, these two compounds are therefore the most likely to be potential GLP-1 receptor agonists. The residues (Asp198 and Tyr148) that interact with the positive control, GF2 and GF3 can be considered as markers of important areas for GLP-1 receptor activation.

## 3. Discussion

In our cytotoxicity study, EGCG, *Camellia sinensis* and RCM-107 dramatically reduced the cell viability of NCI-H716 cells in a dose-dependent manner to around 10%**–**15%. On the other hand, *Alismatis rhizome* showed the least toxicity with 99% viability at the highest concentration (1 mg/mL). 

*Gardeniae fructus* became cytotoxic to the cells at a lower concentration and steadily decreased the cell viability down to 60% viability at the highest concentration. A recent in vivo study reported that administering *Gardeniae fructus* at medium high and high concentration (equivalent to 10 g/day/person and 20 g/day/person) caused GI injury (worse in small intestine and colon than stomach), such as mucus, white cells, and red cells present in stool or inflammation in GI tract [15]. This finding correlates with our cell viability results of the colorectral cells treated with *Gardeniae fructus*.

Furthermore, it is important to note that the cytotoxicity tests of RCM-107 and the individual herbs on NCI-H716 showed an enhancement of cell viability at low concentration before they became cytotoxic. One possible explanation is hormesis, a dose response characterized as stimulation by a low-dose treatment, whereas inhibition by a high-dose treatment can be seen [16].

In the GLP-1 secretion tests, the same concentration of RCM-107, *Camellia sinensis*, *Gardeniae fructus* and EGCG were used based on the cell viability results, while higher concentrations of the other single herbs were used because they were less cytotoxic. Only concentrations with cell viability that exceeded 96% were selected in the experiments. The single herb *Gardeniae fructus* that contained multiple compounds was found to have greater effects on increasing GLP-1 secretion than the natural product EGCG. Even though *Sophorae flos* presented higher protein secretion level, the concentration that could be used in the experiment was higher than *Gardeniae fructus*. Therefore, it can be proposed that *Gardeniae fructus* has higher potential to induce satiety among all single herbs in RCM-107. In fact, geniposide, a bioactive chemical constituent form, has already been reported as a GLP-1 receptor agonist [10,17,18]. 

However, the RCM-107 formula, containing eight ingredients at 3:1(*w/w*) ratio with *Camellia sinensis* being three-fold higher in concentration than the other seven herbs (a constant value), did not significantly stimulate GLP-1 secretion. There are two possible reasons for this result: (1) the highest concentration (had 96% cell viability or more) of RCM-107 that can be used in the ELISA experiments was too low to lead to a statistically significant effect; (2) although *Gardeniae fructus* had secreted higher level of GLP-1 than the positive control EGCG, the small proportion of this individual herb in RCM-107 may have limited the effect of this whole formula on releasing GLP-1 hormone.

In order to study the mechanisms of action of *Gardeniae fructus* for GLP-1 stimulation and propose the key compounds for GLP-1 receptor activation, the ligand–receptor interactions that are critical for agonist-induced receptor have been predicted using molecular docking. The structure described by PDBID 5NX2 is an active GLP-1 receptor structure that has been used to study agonist binding and receptor activation [19]. Amino acid residues include Lys197, Asp198, Lys202, Asp215, Arg227 and Lys288, together with Met204, Tyr205 and Trp306, were reported to be essential for GLP-1 receptor activation [20]. Specifically, Asp198 within the binding pocket is well known for its vital role in agonist binding and receptor activation [21]. Additionally, residue Tyr148 plays a crucial role in peptide agonist affinity [22]. According to the results, both GF2 and GF3 formed hydrogen bonding interactions with Asp198 and Tyr148, suggesting they may be capable to act as GLP-1 receptor agonists. In addition, GF3 formed hydrogen bonds with Lys197, Lys202, Asp198 and Tyr148, and hydrophobic interactions with Tyr 205, and all residues have been considered important for receptor activation. This may indicate that GF3 may be a more potent agonist than EGCG, for which hydrogen bonding interactions were only predicted to be formed with three of the important receptor binding residues, namely, Tyr148, Asp198 and Tyr205. Therefore, GF2 and GF3 are proposed to be potential GLP-1 receptor agonists, and the presence of Asp198 and Tyr148 may be important for those two compounds to act as agonists. 

Taking all of our data together, we can suggest that RCM-107 could be optimized via increasing the concentration of *Gardeniae fructus*. Considering the potential intolerances that might be caused by this herb, the maximum concentration needs to be carefully identified. As the Chinese Pharmacopoeia (Edition 2015) states, the safe dose range of *Gardeniae fructus* is 6-10 g/day/person [15]. Currently, 0.05 g of extracted *Gardeniae fructus* (0.5 g of dry content) is used in one RCM-107 capsule, and the suggested daily dosage according to the Therapeutic Goods Administration (TGA) is two capsules, three times a day. Hence, the proportion of *Gardeniae fructus* can be increased to 0.1 g (1 g of dry herb) per capsule to minimize the potential adverse effects. In addition, the chemical compounds GF2 3-epioleanolic acid and GF3 crocin from *Gardeniae fructus* could potentially act as GLP-1 receptor agonists, although further validations are required. 

## 4. Materials and Methods 

### 4.1. Materials 

Dulbecco’s modified Eagle’s medium (DMEM, 11885084), RPMI 1640 (11875085), Sodium Pyruvate (100 mM, 11360070), phosphate buffered saline (PBS, 10010023), Geltrex™ hESC-Qualified, Ready-To-Use, Reduced Growth Factor Basement Membrane Matrix (A1569601), TrypLE™ EXPRESS (12604013) and Penicillin-Streptomycin (15070063) were from Life Technologies (Mulgrave, Victoria, Australia). Fetal bovine serum (FBS, 12003C), the reagent cell counting kit-8 (CCK-8, 96992), bovine serum albumins (A9418), KREBS-ringers bicarbonate buffer (K4002) and EGCG (E4268) were from Sigma-Aldrich ( Castle hill, New South Wales, Australia). Glucagon-like peptide-1 (GLP-1) Active ELISA Kit (EGLP-35K) were obtained from Abacus dx/ Millipore (Meadowbrook, Queensland, Australia). The RCM-107 capsules (15733) were provided by Tong Kang Lee Chinese medicine clinic (Kensington, Victoria, Australia). Individual herbal granules (Nong’s, HK) consisting of *Camellia sinensis*/matcha/Lu cha ye; *Poria/Poria cocos (schw.) Wolf.*/Fu ling (A1601642); *Nelumbinis folium*/*Nelumbo nucifera Gaertn.*/He ye (A1601450); *Alismatis rhizome*/*Alisma orientalis (sam.) Juzep.*/Ze xie (A1600004); *Plantaginis semen*/*Plantago asiatica L.*/Che qian zi (A1600813); *Cassiae semen*/*Cassia obtusifolia L./*Jue ming zi (A1500060); *Sophorae flos*/*Sophora japonica L.*/Huai hua (A1601428) and *Gardeniae fructus*/*Gardenia jasminoides Ellis.*/Zhi zi (A1600810) were supplied by GL natural health care clinic (Strathmore, Victoria, Australia).

### 4.2. Sample Preparation

Each RCM-107 capsule contains 500 mg extracted herbal powder (including 150 mg of *Camellia sinensis* and 50 mg of the rest seven ingredients). Herbal powder and granules were obtained by water extraction of the raw materials via boiling and lyophilisation. The ratio between extracted: crude/dry herbs is 1:10. For cell viability test, thirty six milligrams of the RCM-107, and its eight individual ingredients were mixed with 3 mL of DMEM (the positive control EGCG was dissolved in PBS), vortexed and sonicated for 20 min, then filtered by Millex-HP 0.22 µm filters (Millipore, Bayswater, Victoria, Australia). All samples were serial diluted to obtain 5 or 6 calibration points (final concentration 0.05**–**1 mg/mL or 0.005**–**1 mg/mL). Prior to the ELISA experiment, 33 mg of all samples were dissolved in Krebs–Ringer bicarbonate buffer, vortexed and sonicated for 20 min then filtered by Millex-HP 0.22 µm filters (Millipore, Bayswater, Victoria, Australia), before they have been diluted to the desired concentration. 

### 4.3. Cell Culture

Human NCI-H716 cells (the American Type Culture Collection; ATCC, Manassas, VA, USA) was maintained in RPMI 1640 supplemented with 10% (*v/v*) FBS, 1 mM sodium pyruvate, 100 µg/mL penicillin and 100 µg/mL streptomycin and incubated at 37 °C with 5% CO_2_.

### 4.4. Cell Viability Assay

Cell viability was determined by colorimetric assay using a cell counting kit-8 (CCK-8; Sigma-Aldrich, Castle hill, New South Wales, Australia), following the instructions provided by the manufacturer. A hundred microliter of NCI-H716 cells (5 × 10^3^ cells/well) were seeded per well into a 96 well-plate in DMEM complete media and incubated at 37 °C with 5% CO_2_. After 24 h, RCM-107, its eight individual herbal ingredients and EGCG (0.005**–**1 mg/mL) were added to cells; the cells with medium only (without treatments) served as the negative control. After another 24-h incubation, 10 µL of CCK-8 solution was added to each well. After further incubation for 3 h at 37 °C, the optical absorption was measured at 450 nm using CLARIOstar^®^ microplate reader (BMG labtech, Mornington, Victoria, Australia).

Cell viability was calculated following the equation:Cell viability = (A_sample_**−** A_sample blank_)/(A_control_**−** A_control blank_) × 100% 
where A indicates the absorbance of each well. A_sample_ and A_sample blank_ represent absorbance values of the sample solution with or without cells, while A_control_ and A_control blank_ are absorbance values of the control (contains no tested samples) with or without cells, respectively.

### 4.5. GLP-1 Secretion ELISA

Cells in DMEM were plated at 1 × 10^6^ cells/well in 12-well microplate coated with matrix and incubated for 48 h. Then, the test samples prepared in Krebs–Ringer bicarbonate buffer (KRB, 128.8 mmol/L NaCl, 4.8 mmol/L KCl, 1.2 mmol/L KH_2_PO_4_, 1.2 mmol/L MgSO_4_, 2.5 mmol/L CaCl_2_, 5 mmol/L NaHCO_3_, 10 mmol/L HEPES, and 0.2% BSA (*w/v*), pH 7.4) [12] were added to each well followed by 2 h incubation at 37 °C [23]. The supernatant was collected and centrifuged for 5 min at 1300 rpm and immediately assayed via a GLP-1 active ELISA kit, following the instruction provided by the manufacturer. The active GLP-1 contents were normalized with the protein concentrations corresponding to each sample well and expressed as ng/mL.

### 4.6. Molecular Docking

A total of 22 ligands from *Gardeniae fructus* were obtained from the literature, including its major bioactive chemical constituents [24] and compounds that met the common requirements for potential favorable therapeutic efficacy, such as good oral bioavailability (OB) ≥ 30%, drug likeness (DL) ≥ 0.18 and intestinal epithelial permeability (Caco-2) ≥ **−**0.40 [25,26].

Autodock Vina (together with the graphical frontend PyRx version 0.8) was used for all docking calculations (https://pyrx.sourceforge.io/). The structures of small molecules (herbal ligands, the positive control EGCG) and macromolecules (the human GLP-1 receptor x-ray structure PDB entry: 5NX2) required to initiate structure-based virtual screening were obtained and preprocessed. The structures of small molecules were acquired from Pubchem (https://pubchem.ncbi.nlm.nih.gov/) or Traditional Chinese Medicine Systems Pharmacology (TCMSP) (http://tcmspw.com/tcmsp.php) as Standard Database Format (SDF) files, and they were then translated to Protein Data Bank (PDB) files, which can be recognized by PyRx. The 3D protein structures were obtained from RCSB Protein Databank (www.rcsb.org) and modified using Visual Molecular Dynamics (VMD) by removing water molecules to obtain protein-only structures.

Subsequently, both selected protein (5NX2) and ligand files were loaded to PyRx as macromolecules and ligand, respectively. Proteins were fixed, while ligands could have rotatable torsions. Protein and ligand hydrogens were automatically added using the PyRx hydrogen (H) repair functionality. Autodock Vina automatically samples different conformations of the ligands to best fit the predicted binding site. Using the atomic coordinates in the PDB file of 5NX2, the center of the box was defined as being located at the approximate center of the protein, with coordinates X: **−**15.864; Y: 23.3464; Z: 5.5536. The dimensions (angstrom) were X: 65.8783; Y: 63.5080; and Z: 91.8356, and the exhaustiveness parameter was set to 64 for all dockings. The 2D ligand and target interaction diagram can be generated by Discovery Studio Visualizer (DSV) Version 4.5, which can be found from BIOVIA, San Diego, CA, USA (https://www.3dsbiovia.com/products/collaborative-science/biovia-discovery-studio/). The 3D image and binding sites were also observed and displayed in DSV. The top three ligands with higher binding affinity from *Gardeniae fructus* have been selected for inter-residue interaction analysis, compared to that of the positive control EGCG.

### 4.7. Statistical Analyses

Samples and controls were all run in triplicates. Results were presented as mean ± standard deviation (SD) from three independent experiments. One-way analysis of variance (ANOVA) and two-way ANOVA followed by Tukey’s test were used for statistical analysis via Prism 8.0 (GraphPad Software, San Diego, CA, USA).

## 5. Conclusions

In summary, *Camellia sinensis* and RCM-107 were more cytotoxic to NCI-H716 cells than the other single herbs present in this herbal formula, while *Alismatis rhizome* showed nonsignificant cytotoxicity at the highest concentration. Even though RCM-107 did not show a significant increase in GLP-1 secretion, an individual herb present in this formula, *Gardeniae fructus*, demonstrated significantly greater effects on inducing GLP-1 secretion than the natural product EGCG. Thus, we propose that RCM-107 can be optimised in regards to enhanced satiety and appetite suppression by increasing the proportion of *Gardeniae fructus*. Furthermore, 3-epioleanolic acid and crocin from *Gardeniae fructus* are likely to be the key compounds that can potentially lead to a stimulation of GLP-1 secretion. The residues Asp198 and Tyr148 are proposed to be vital for GLP-1 receptor agonist binding. In the near future, in vivo studies that investigate the effects on GLP-1 secretion of the individual compounds crocin and 3-epioleanolic acid, compared with the single herb *Gardeniae fructus*, as well as the optimized form of RCM-107 formula, will need to be conducted.

## Figures and Tables

**Figure 1 ijms-21-02854-f001:**
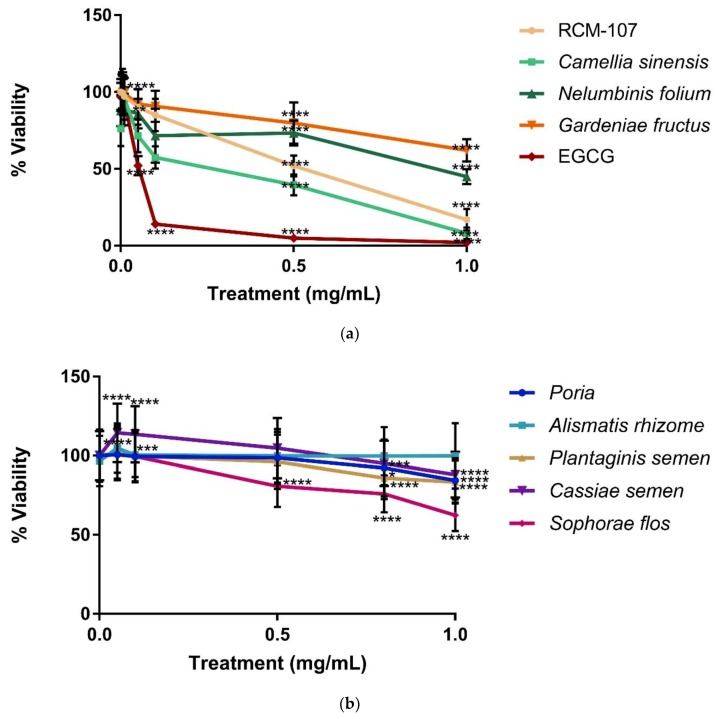
(**a**) Cell viability after 24 h of treatment of NCI-H716 cells with RCM-107, Camellia sinensis, Nelumbinis folium, Gardeniae fructus and epigallocatechin gallate (EGCG). Values are mean ± SD of three replicates per experiment ** *p* < 0.01 and **** *p* < 0.0001 when compared with the corresponding untreated cells.; (**b**) Cell viability after 24 h of treatment of NCI-H716 cells with Poria, Alismatis rhizome, Plantaginis semen, Cassiae semen and Sophorae flos. Values are mean ± SD of three replicates per experiment * *p* < 0.05, *** *p* < 0.005 and **** *p* < 0.0001 when compared with the corresponding untreated cells.

**Figure 2 ijms-21-02854-f002:**
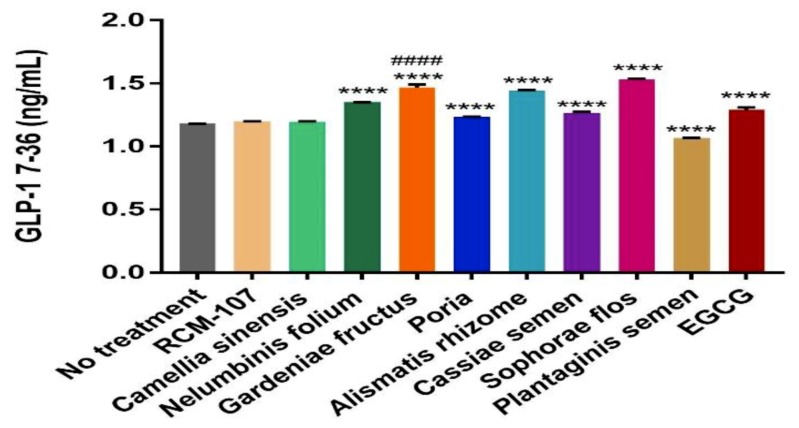
Effects of RCM107, its eight ingredients and EGCG on glucagon-like peptide-1 (GLP-1) release from the NCl-H716 cells. The maximal concentrations used for ELISA were selected from the highest concentration that maintained cell viability above 96% (0.01 mg/mL of RCM-107, Camellia sinensis, Gardeniae fructus and EGCG; 0.05 mg/mL of Nelumbinis folium; 0.1 mg/mL of Sophorae flos; 0.5 mg/mL of Plantaginis semen; 0.8 mg/mL of Poria and Cassiae semen; and 1 mg/mL Alismatis rhizome were used). Cells were incubated for 2 h with test samples and the active GLP-1 contents were normalized and expressed as ng/mL. Values are means ± SD of three replicate experiments. **** *p* < 0.0001 vs cells with no treatments; #### *p* < 0.0001 vs the positive control EGCG.

**Figure 3 ijms-21-02854-f003:**
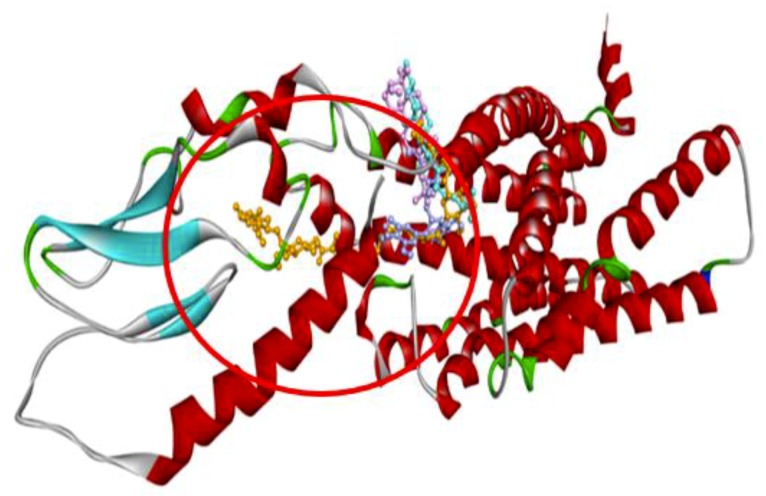
3D ribbon diagram showing the binding site of predicted leading compounds stigmasterol (blue), 3-epioleanolic acid (pink), crocin (orange) and the positive control EGCG (lilac), on the GLP-1 receptor (PDBID: 5NX2).

**Figure 4 ijms-21-02854-f004:**
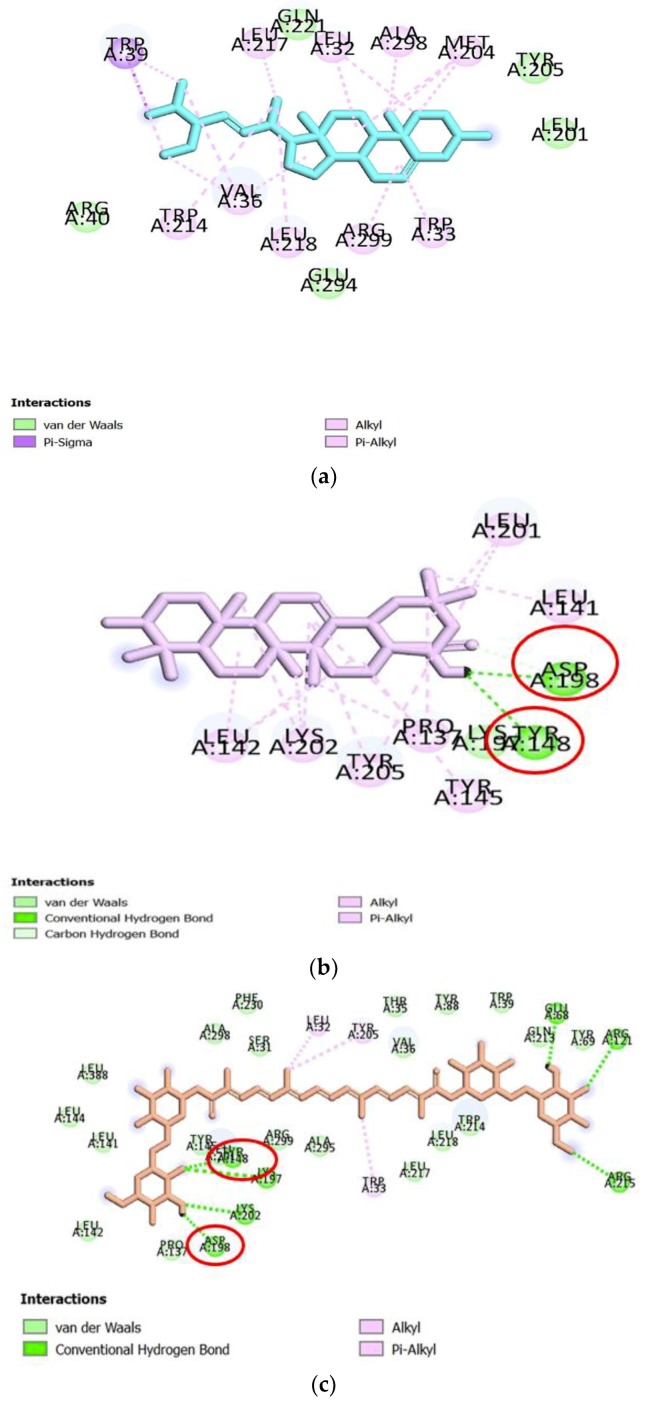
2D diagram of top three predicted ligands from *Gardeniae fructus* and the positive control with 5NX2. Hydrogen bonds (green circles with traced lines), π-Alkyl interactions (pink circles with traced lines) and π-Sigma interaction (purple circles with traced lines). (**1**) Stigmasterol, (**2**) 3-epioleanolic acid, (**3**) crocin and (**4**) EGCG. The key residues in common between EGCG and the three ligands have been circled in red and discussed in the text.

**Table 1 ijms-21-02854-t001:** Binding affinity of each selected *Gardeniae fructus* ligand to 5NX2. The binding affinity has been coloured with darker blue, indicating the most negative values. Three ligands highlighted in red have been selected and discussed.

Ligand ID	Ligand Name	Molecular Weight (g/mol)	Binding Affinity (kcal/mol)
**GF1**	stigmasterol	412.7	−10.9
**GF2**	3-epioleanolic acid	456.7	−10.8
**GF3**	crocin	977	−10.7
**GF4**	sudan III	352.4	−10.2
**GF5**	supraene	401.8	−8.7
**GF6**	3-methylkempferol	300.3	−8.4
**GF7**	isoimperatorin	270.3	−8.3
**GF8**	ammidin	270.3	−8.3
**GF9**	shanzhiside	392.35	−8.2
**GF10**	corymbosin	358.3	−8.2
**GF11**	scandoside methyl ester	404.4	−8
**GF12**	chlorogenic acid	354.31	−8
**GF13**	crocetin	328.4	−8
**GF14**	gardoside	374.34	−7.9
**GF15**	picrocrocinic acid	346.37	−7.9
**GF16**	geniposide	388.41	−7.7
**GF17**	geniposidic acid	374.38	−7.4
**GF18**	gardenoside	404.46	−7.3
**GF19**	gardenogenin A	242.22	−6.8
**GF20**	genipin	226.3	−6.6
**GF21**	ethyl oleate (NF)	310.6	−5.9
**GF22**	mandenol	308.6	−5.9

**Table 2 ijms-21-02854-t002:** Top 3 ligands predicted from *Gardeniae fructus* with the strongest binding affinity and their interactions with 5NX2.

Herbal Names	Ligands	PubChem ID	Molecular Formula	Molecular Weight (g/mol)	Hydrogen Bond (Amino Acids)
***Gardeniae fructus***	stigmasterol (GF1)	5280794	C_29_H_48_O	412.7	N/A
***Gardeniae fructus***	3-epioleanolic acid (GF2)	11869658	C_30_H_48_O_3_	456.7	**Asp198** **Tyr148**
***Gardeniae fructus***	crocin (GF3)	5281233	C_44_H_64_O_24_	977	**Tyr148** Lys197 Lys202 **Asp198** Glu68 Arg121 Arg215
**The Positive control**	EGCG	65064	C_22_H_18_O_11_	458.4	Tyr148 Asp198 Ser31 Tyr205

Notes: Their common residues shared with EGCG have been marked in bold.

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
