# Peer review of "The Effects of a Weight-Loss Herbal Formula RCM-107 and Its Eight Individual Ingredients on Glucagon-Like Peptide-1 Secretion—An In Vitro and In Silico Study"

_ijms, 2020, doi:10.3390/ijms21082854_

Round 1
Reviewer 1 Report
The manuscript shows the in vitro effects of a weight loss herbal formula RCM-107 and its eight individual ingredients on glucagon-like peptide-1 secretion. Additionally, to study the mechanisms of action of Gardeniae fructus for GLP-1 stimulation and to propose the key compounds for GLP-1 receptor activation some molecular docking studies were employed.
In my opinion the study is valuable and the presented results are convincing. The paper is concise and well written. There are only six minor points which should be addressed by the authors:
- Quality of Fig. 1 and 4 should be improved
- Abbreviations should be defined at first mention and used consistently thereafter (e.g. EGCG)
- The explanation of the cell line NCI-H716 should be given at first mention
- English short form (e.g. it’s) should not be used within the paper
- Lines 301 and 312: Number of cells seeded: 5x103 cells/ well and 1x106 cells/well vs. 5x103 cells/ well and 1x106 cells/well
- References should be carefully checked and unified (e.g. position 2)
Author Response
Comment 1:
Quality of Fig. 1 and 4 should be improved
Response 1:
The resolutions of Fig. 1 and 4 have been improved to 1200 DPI (12212 pixels width and 3080 pixels height) and 1778 DPI (10248 pixels width and 9902 pixels height) respectively. Also, these have been enlarged in the text.
Comment 2:
Abbreviations should be defined at first mention and used consistently thereafter (e.g. EGCG)
Response 2:
Line 56: Epigallocatechin gallate (EGCG) has been defined where it first appeared.
Lines 146 and 147: Protein Data Bank identification code (PDBID) has been defined at first mention.
Lines 151 and 152: Three-dimensional (3D) two-dimensional (2D) have been defined when first mentioned.
Line 268: Phosphate buffered saline (PBS) has been added to where it was first mentioned.
Lines 331 and 332: The full names of the followings have been added in the manuscript: Traditional Chinese Medicine Systems Pharmacology (TCMSP); Standard Database Format (SDF); files Protein Data Bank (PDB)
Line 376: A list of abbreviations used in the text has been added.
Comment 3:
The explanation of the cell line NCI-H716 should be given at first mention
Response 3:
The description of the cell line NCI-716 has been moved from the discussion to the introduction (lines 75-80), where it was mentioned first.
Comment 4:
English short form (e.g. it’s) should not be used within the paper
Response 4:
Line 215: the English short form it’s has been changed to it is.
Line 299: following manufacturer’s instructions has been changed to following the instructions provided by the manufacturer.
Comment 5:
Lines 301 and 312: Number of cells seeded: 5x103 cells/ well and 1x106 cells/well vs. 5x103 cells/ well and 1x106 cells/well
Response 5:
Lines 300 and 313: superscripts have been corrected to 5x103 cells/ well and 1x106 cells/well.
Comment 6:
References should be carefully checked and unified (e.g. position 2)
Response 6:
References have been checked one by one according to IJMS reference style requirements via endnotes and also rechecked in the text. They have been unified:
Corrected some of the journal title abbreviations,
Ensured the author names are consistent.

Reviewer 2 Report
- The quality of figures of the manuscript must be improved. I could not check Figure 1 and Figure 4 due to poor qualities.
- The authors should performed the docking with the protein 5NX2, which is bound to a truncated peptide. The authors should mention the exact coordinate of binding site used for docking as well as grid size.
- In section 4.4, company name of 'Cell Counting Kit-8 (CCK-8) colorimetric assay' should be provided.
- Throughout the manuscript subsripts are not found (eg CO2, KH2PO4). Line 301, '5x103 cells/well' should be replaced with '5x103 cells/well'.
- Did the authors used any standard compound in cell viability assay performed here? If yes, mention in the manuscript.
Author Response
Comment 1
The quality of figures of the manuscript must be improved. I could not check Figure 1 and Figure 4 due to poor qualities.
Response 1
The resolutions of Fig. 1 and 4 have been improved to 1200 DPI (12212 pixels width and 3080 pixels height) and 1778 DPI (10248 pixels width and 9902 pixels height) respectively. Also, these have been enlarged in the text.
Comment 2
The authors should performed the docking with the protein 5NX2, which is bound to a truncated peptide. The authors should mention the exact coordinate of binding site used for docking as well as grid size.
Response 2
The following has been added to the methods section, section 4.6 lines 340-344 to address this comment:
Using the atomic coordinates in the PDB file of 5NX2, the center of the box was defined as being located at the approximate center of the protein, with coordinates X: -15.864; Y: 23.3464; Z: 5.5536. The dimensions (angstrom) were X: 65.8783; Y: 63.5080; Z: 91.8356 and the exhaustiveness parameter was set to 64 for all dockings.
Comment 3
In section 4.4, company name of 'Cell Counting Kit-8 (CCK-8) colorimetric assay' should be provided.
Response 3
In the methods, section 4.4, lines 297 and 298, the company name of CCK-8 has now been provided: Cell viability was determined by colorimetric assay using a cell counting kit-8 ( CCK-8; Sigma-Aldrich, Australia)
Comment 4
Throughout the manuscript subsripts are not found (eg CO2, KH2PO4). Line 301, '5x103 cells/well' should be replaced with '5x103 cells/well'.
Response 4
The manuscript has been carefully checked again and subscripts or superscripts have been corrected where required:
Lines 295 and 301: 5% CO2 ;
Lines 315 and 316: KH2PO4, MgSO4, CaCl2, NaHCO3;
Lines 300 and 313: 5x103 cells/ well and 1x106 cells/well.
Comment 5
Did the authors used any standard compound in cell viability assay performed here? If yes, mention in the manuscript.
Response 5
To best of our knowledge, there is no standard toxic compound that is used for this particular cell line. This is reflected in the literature where this cell line has been used to conduct cell viability assays. It is typical that such studies include negative controls for cytotoxicity comparison [1-4], as we also have included in this study (a negative control of normal cells with medium only). This additional detail has now been mentioned in the method part of the manuscript (section 4.4):
Lines 301-303 After 24 hours, RCM-107, its eight individual herbal ingredients and EGCG (0.005-1 mg/mL) were added to cells; the cells with medium only (without treatments) served as the negative control.
Reviewer 3 Report
It is not possible to read the pictures.
Authors compare extracts at very different concentrations, some of them too high to be reached physiologically.
They mix active GLP1 with total GLP1
Author Response
Comment 1:
It is not possible to read the pictures.
Response 1:
The resolutions of Fig. 1 and 4 have been improved to 1200 DPI (12212 pixels width and 3080 pixels height) and 1778 DPI (10248 pixels width and 9902 pixels height) respectively. Also, these have been enlarged in the text, so they can be readable.
Comment 2:
Authors compare extracts at very different concentrations, some of them too high to be reached physiologically.
Response 2:
As mentioned in the text, section 2.2, lines 115-116, the maximal treatment concentrations used for the ELISA were selected from the highest concentration that maintained cell viability above 96% (ie. The highest tolerable dose). We tried to use the highest non-cytotoxic concentration to maximise the chances of observing GLP-1 stimulating effects of the extracts, which may have been missed at lower concentrations.
Following this methodology, Gardeniae fructus, Camellia sinensis and RCM-107 were compared to the positive control EGCG as they had similar cell viability as per the assay results and the concentration used in the ELISA were the same (10µg/mL) (mentioned in lines 126-128).
Sophorae flos was compared with Gardeniae fructus at different maximum concentrations. The differences in their concentrations have been described in the text, lines 129-130:
Even though Sophorae flos exhibited the highest protein secretion level, the concentration used in the experiment was 10 times higher than Gardeniae fructus due to its lower cytotoxicity to NCI-H716 cells.
The purpose of these in vitro studies was not to create a physiological model in the cell culture system, but rather to test whether GLP-1 stimulation could be achieved through maximal stimulation of each extract in cell culture. In such studies, we must always keep in mind that the physiological concentration maxima will be far lower and therefore positive effects may be lessened in vivo.
Comment 3:
They mix active GLP1 with total GLP1
Response 3:
We are not quite sure which section the reviewer is referring to. However, we have clarified the text to improve understanding.
ELISA was used to measure the active GLP-1, so we have changed:
Section 4.5 line 319: The active GLP-1 contents were normalized
Figure 2 caption, line 141: The active GLP-1 contents were normalized

Round 2
Reviewer 2 Report
The authors used 2D interaction diagram Discovery Studio Visualizer (DSV) for Figure 4. Right click on the 2D diagram in DSV and go to 'Atom', Instead of default 'line', select 'stick'.
Additionally, the author may increase the font size of amino acid residues from 'General', increase 'Atom/Residue Font and Scale' to 2.5.
Author Response
Comment 1
The authors used 2D interaction diagram Discovery Studio Visualizer (DSV) for Figure 4. Right click on the 2D diagram in DSV and go to 'Atom', Instead of default 'line', select 'stick'.
Additionally, the author may increase the font size of amino acid residues from 'General', increase 'Atom/Residue Font and Scale' to 2.5.
Response 1
Figure 4.1-4.4 lines 183-190: Display Style has been revised.
1) Selected “atom” then “stick”;
2) Increased the front size from “Atom/residue front and scale” to 2.5.
All figures have been cropped slightly to fit in the text better. Resolutions have been improved again (1778DPI, 10384 pixels width and 9582 pixels height).
Reviewer 3 Report
nothing to add
Author Response
Comment:
Nothing to add
Response
All authors proof read the manuscript again and figures have been checked and adjusted accordingly.